# Using ΔK280 Tau_RD_ Folding Reporter Cells to Screen TRKB Agonists as Alzheimer’s Disease Treatment Strategy

**DOI:** 10.3390/biom13020219

**Published:** 2023-01-23

**Authors:** Zheng-Kui Weng, Te-Hsien Lin, Kuo-Hsuan Chang, Ya-Jen Chiu, Chih-Hsin Lin, Pei-Hsuan Tseng, Ying-Chieh Sun, Wenwei Lin, Guey-Jen Lee-Chen, Chiung-Mei Chen

**Affiliations:** 1Department of Life Science, National Taiwan Normal University, Taipei 106, Taiwan; 2Department of Neurology, Chang Gung Memorial Hospital, School of Medicine, Chang-Gung University, Taoyuan 333, Taiwan; 3Department of Chemistry, National Taiwan Normal University, Taipei 106, Taiwan

**Keywords:** Alzheimer’s disease, BDNF-TRKB signaling, TRKB agonist, heterocyclic compound, Tau cell model

## Abstract

Misfolded aggregation of the hyperphosphorylated microtubule binding protein Tau in the brain is a pathological hallmark of Alzheimer’s disease (AD). Tau aggregation downregulates brain-derived neurotrophic factor (BDNF)/tropomycin receptor kinase B (TRKB) signaling and leads to neurotoxicity. Therefore, enhancement of BDNF/TRKB signaling could be a strategy to alleviate Tau neurotoxicity. In this study, eight compounds were evaluated for the potential of inhibiting Tau misfolding in human neuroblastoma SH-SY5Y cells expressing the pro-aggregator Tau folding reporter (ΔK280 Tau_RD_-DsRed). Among them, coumarin derivative ZN-015 and quinoline derivatives VB-030 and VB-037 displayed chemical chaperone activity to reduce ΔK280 Tau_RD_ aggregation and promote neurite outgrowth. Studies of TRKB signaling revealed that ZN-015, VB-030 and VB-037 treatments significantly increased phosphorylation of TRKB and downstream Ca^2+^/calmodulin-dependent protein kinase II (CaMKII), extracellular signal-regulated kinase 1/2 (ERK) and AKT serine/threonine kinase (AKT), to activate ribosomal S6 kinase (RSK) and cAMP response element-binding protein (CREB). Subsequently, p-CREB enhanced the transcription of pro-survival BDNF and BCL2 apoptosis regulator (BCL2), accompanied with reduced expression of anti-survival BCL2-associated X protein (BAX) in ΔK280 Tau_RD_-DsRed-expressing cells. The neurite outgrowth promotion effect of ZN-015, VB-030 and VB-037 was counteracted by a RNA interference-mediated knockdown of TRKB, suggesting the role of these compounds acting as TRKB agonists. Tryptophan fluorescence quenching analysis showed that ZN-015, VB-030 and VB-037 interacted directly with a *Pichia pastoris*-expressed TRKB extracellular domain, indirectly supporting the role through TRKB signaling. The results of up-regulation in TRKB signaling open up the therapeutic potentials of ZN-015, VB-030 and VB-037 for AD.

## 1. Introduction

A number of neurodegenerative diseases, including Alzheimer’s disease (AD), progressive supranuclear palsy, corticobasal syndrome and frontotemporal dementia, are characterized by accumulation of β-sheet-rich misfolded Tau proteins [1]. Encoded by *MAPT*, Tau participates in microtubule dynamics and assembly, and also plays a role in axonal transport [2,3,4]. Various genetic mutations have been attributed to inducing Tau protein misfolding. For example, deletion of a highly conserved lysine 280 (∆K280) in the conserved 18-amino acid repeat domain of Tau (Tau_RD_) was found in patients with tauopathies [5,6]. Overexpression of ∆K280 Tau_RD_ in HEK293T cells increased the formation of misfolded aggregates, suggesting its potential to accelerate Tau misfolding [7]. Of note, the misfolded Tau aggregation down-regulates brain-derived neurotrophic factor (BDNF) signaling pathways [8,9], implying the potential involvement of BDNF in the pathogenesis of Tau-mediated neurodegeneration.

BDNF is a neurotrophic factor involving neuronal growth and survival [10]. Postmortem studies demonstrate reduced expression of BDNF in hippocampus, cortex and basal nucleus of Meynert in patients with AD, even in the pre-clinical stages [11,12]. The binding of BDNF to tropomyosin-related kinase B (TRKB) induces autophosphorylation, and then activates phosphoinositide 3-kinase (PI3K)-AKT serine/threonine kinase (AKT), extracellular signal-regulated kinase (ERK) and phospholipase-C-γ (PLC-γ) pathways [13]. The PI3K-AKT pathway suppresses cell apoptosis by reducing the translocation of apoptotic B-cell lymphoma 2 (BCL2)-associated X protein (BAX) from the cytoplasm to mitochondria [14]. The 90-kDa ribosomal S6 kinase (RSK) involved in cell survival is a downstream effector of ERK and both ERK and RSK phosphorylate the transcription factor cAMP responsive element-binding protein 1 (CREB) [15]. Activation of CREB promotes transcription of the BDNF neurotrophin [16] and BCL2 anti-apoptosis regulator [17], to regulate neuronal growth and differentiation, synaptic plasticity, spatial memory and long-term memory formation, as well as to ensure neuronal survival [18]. The PLC-γ pathway also mediates CREB phosphorylation through Ca^2+^ influx in cortical neurons [19], and decreased phosphorylation of Ca^2+^/calmodulin-dependent protein kinase II (CaMKII) has been observed in SH-SY5Y cells expressing pro-aggregator Tau [20]. In AD animal models, overexpression of BDNF exerts the potential to rescue deficits in learning and memory [21,22].

Currently, a treatment to halt neurodegeneration in AD is urgently needed. Flavones, benzofurans, coumarins and quinolines are heterocyclic compounds with broad ranges of biological activities, such as antioxidant, anti-dementia and anti-inflammatory properties [23,24,25]. Although a few TRKB agonists such as flavone 7,8-DHF (7,8-dihydroxy flavone) have demonstrated neuroprotective potential against amyloid β (Aβ)-mediated neurodegeneration [26,27], the potential of TRKB agonists in Tau-mediated neurodegeneration remains to be determined. Previously, we found the effects of coumarin derivatives LMDS-1 and -2 against Tau-mediated neurotoxicity by activating TRKB signaling [8]. We have also shown that coumarin derivatives ZN-014 and ZN-015 reduced Aβ neurotoxicity via pleiotropic neuroprotective mechanisms [28], and quinoline derivatives VB-030 and VB-037 reduced Tau neurotoxicity through inhibiting glycogen synthase kinase-3β (GSK-3β) kinase activity and/or p-P38 (T180/Y182) [29]. To expand the capacity of heterocyclic compounds as TRKB agonists, we examined the neuroprotective potentials of ZN-006 (flavone), -013 (benzofuran), -014 and -015 (coumarin), and VB-030, -031, -037 and -041 (quinoline) in ΔK280 Tau_RD_-DsRed folding reporter SH-SY5Y cells and whether they worked as TRKB agonists to provide neuroprotection. The binding affinity of potential TRKB agonists to the extracellular domain of TRKB (TRKB-ECD) was also assessed by tryptophan fluorescence quenching.

## 2. Materials and Methods

### 2.1. Compounds

One flavone ZN-006 [2-((5-hydroxy-4-oxo-2-phenyl-4H-chromen-7-yl)oxy)acetic acid], one benzofuran ZN-013 [(4-methoxybenzofuran-2-yl)(phenyl)methanone], and four quinolines VB-030 [2-(pyridin-4-yl)-4-(p-tolyl)quinoline], VB-031 [(E)-3-(3-(1H-benzo[d]imidazol-2-yl)acryloyl)-6-methyl-4-phenylquinolin-2(1H)-one], VB-037 [(E)-4-(3-(2-(5-nitroquinolin-2-yl)vinyl)quinolin-2-yl)morpholine] and VB-041 [2-(pyridin-4-yl)-N-(3-(N-(3,4,5,6-tetrahydro-2H-azepin-7-yl)sulfamoyl)phenyl)quinoline-4-carboxamide] were purchased from Enamine (Kyiv, Ukraine). Procedures for producing coumarins ZN-014 (ethyl 5-hydroxy-2-oxo- 2H-chromene-3-carboxylate) and ZN-015 [(*E*)-4-hydroxy-3-(3-(2-hydroxyphenyl)acryloyl)-2H-chromen-2-one] were as stated [28]. Congo red, kaempferol and 7,8-DHF, controls for cellular and/or biochemical assays were obtained from Sigma-Aldrich (St. Louis, MO, USA). In addition, LM-031 (3-benzoyl-5-hydroxychromen-2-one), a control for evaluating TRKB signaling, was synthesized as stated [30].

### 2.2. Cell Culture and Cell Viability Assay

Human neuroblastoma SH-SY5Y cells with inducible ΔK280 Tau_RD_-DsRed expression [7] were maintained in DMEM/F-12 supplemented with fetal bovine serum (FBS, 10%) (Thermo Scientific, Waltham, MA, USA), penicillin/streptomycin (100 U/mL), selection antibiotics hygromycin (100 μg/mL) and blasticidin (5 μg/mL) (InvivoGen, San Diego, CA, USA), and 1.5 g/l sodium bicarbonate. Cells were cultured at 37 °C in a 5% CO_2_ atmosphere. The cell viability of ZN and VB compounds (concentration range: 0.1–100 μM) was evaluated by a 3-(4,5-dimethylthiazol-2-yl)-2,5-diphenyltetrazolium bromide (MTT; Sigma-Aldrich) colorimetric assay. The absorbance at 570 nm was recorded by a Multiskan™ GO Microplate Spectrophotometer (Thermo Scientific).

### 2.3. Antioxidant Assays

Free-radical scavenging activity of kaempferol (positive control) and ZN/VB compounds (10–160 μM) was measured using a 1,1-diphenyl-2-picrylhydrazyl (DPPH; Sigma-Aldrich) photometric assay as stated [31]. The absorbance at 517 nm was assessed by a Multiskan™ GO Microplate Spectrophotometer (Thermo Scientific). An oxygen radical absorbance capacity (ORAC) assay [32] of kaempferol and ZN/VB compounds (4–100 μM) was conducted according to the manufacturer’s instructions (OxiSelect™; Cell Biolabs, San Diego, CA, USA). The decrease of fluorescence intensity was monitored by a FLx800 microplate fluorescence reader (BioTek Instruments, Winooski, VT, USA). The ORAC activity was quantified as a trolox-equivalent antioxidant capacity.

### 2.4. Biochemical Fluorescence-Based Tau_RD_ Aggregation Assay

The anti-aggregation activity of congo red (positive control) and ZN/VB compounds (5–20 μM) was assessed using *E. coli*-expressed ΔK280 Tau_RD_ protein [33] and thioflavin T (Sigma-Aldrich) fluorescence assay. The trend of ΔK280 Tau_RD_ amyloid aggregate formation depicted with enhanced and red-shifted fluorescence was recorded by a FLx800 microplate fluorescence reader (BioTek Instruments), where the procedure was previously reported in detail [34].

### 2.5. High-Content Analysis of Cellular ΔK280 Tau_RD_-DsRed Aggregation, Reactive Oxygen Species (ROS) and Neurite Outgrowth

ΔK280 Tau_RD_-DsRed SH-SY5Y cells were plated in the presence of 10 μM retinoid acid (RA; Sigma-Aldrich) to promote neuronal differentiation. The next day, cells were added with congo red or ZN/VB compounds (2–10 μM) for 8 h pretreatment, followed by inducing ΔK280 Tau_RD_-DsRed expression with doxycycline (2 μg/mL). On day 8, cell nuclei were stained with Hoechst 33342 (0.1 μg/mL; Thermo Scientific), and cell images acquired using ImageXpress Micro Confocal High-Content Imaging System (Molecular Devices, San Jose, CA, USA) with DAPI (4′,6-diamidino-2-phenylindole, for nuclei) and TRITC (tetramethylrhodamine, for DsRed) fluorescence filter sets. To detect ROS changes, 10 μM compound-treated cells were stained with Hoechst 33342 and 2′,7′-dichlorodihydrofluorescein diacetate (H_2_DCFDA, 10 μM) fluorescein (Invitrogen, Carlsbad, CA, USA), and cell images captured using DAPI and FITC (fluorescein isothiocyanate, for DCF) filters. For assessing differential neurite outgrowth, 10 μM compound-treated cells were fixed (paraformaldehyde, 4%), permeabilized (Triton X-100, 0.3%), and stained with class III β-tubulin (TUBB3, 1:1000) primary antibody (#802001, BioLegend, San Diego, CA, USA), followed by donkey anti-rabbit Alexa Fluor^®^ 555 secondary antibody (1:1000; #A-31572, Invitrogen) and DAPI (0.1 μg/mL; Sigma-Aldrich) stains. The nuclei and neurite images were acquired with DAPI and TRITC fluorescence filters, respectively. All analyses of images were performed using MetaXpress software (Molecular Device).

### 2.6. Real-Time PCR Analysis

The cDNA was synthesized (SuperScript™ III reverse transcriptase; Invitrogen) using total RNA extracted by TRI Reagent™ (Sigma-Aldrich). The real-time quantitative PCR was conducted with primers that were specific to DsRed and hypoxanthine phosphoribosyltransferase 1 (HPRT1, an endogenous control) as described [7] on the StepOnePlus™ Real-Time PCR System (Applied Biosystems, Foster City, CA, USA). The fold change of the expressed Tau-DsRed mRNA was calculated using formula 2^ΔCt^, as ΔC_T_ = C_T_ (HPRT1) − C_T_ (DsRed), where C_T_ indicates the cycle threshold.

### 2.7. Caspase-1/6 and Acetylcholinesterase (AChE) Activity Assays

After being compounds-treated and RA-differentiated, cells were collected for caspase-1/6 and AChE activity measurements. Cells were lysed in lysis buffer with repeated freeze/thaw cycles and protein concentration in cell lysates quantitated. Types of substrates applied to detect each caspase’s activity were: YVAD-AFC for caspase-1 and VEID-AFC for caspase-6 (BioVision, Milpitas, CA, USA). The reaction mixtures were incubated for 1.5 h at 37 °C according to the manufacturer’s instructions. The fluorescence intensities with excitation/emission wavelengths at 400/505 nm were recorded by a FLx800 microplate fluorescence reader (BioTek Instruments). An AChE activity assay was performed according to the manufacturer’s instructions (Sigma-Aldrich), as described in the Ref. [28]. The mixture was incubated for 2–10 min at room temperature, and absorbance at 412 nm wavelength was measured by Multiskan™ GO Microplate Spectrophotometer (Thermo Scientific).

### 2.8. Western Blot Analysis

Cells were collected and protein extracted as described [34]. The extracted protein (20 µg) was mixed with a 6X SDS sample buffer and boiled for 5 min. The denatured protein samples were then separated by 10–12% SDS-polyacrylamide gel and transferred to a polyvinylidene difluoride (PVDF) membrane (Pall Corporation, Port Washington, NY, USA). The blotted membrane was blocked with 3% bovine serum albumin in Tris-buffered saline with 0.1% Tween-20 and probed with primary antibodies against AKT (1:1000; #9272, Cell Signaling Technology (CST), Danvers, MA, USA), p-AKT S473 (1:1000; #4060, CST), BAX (1:1000; #2772, CST), BCL2 (1:1000; #3033, BioVision), BDNF (1:1000; #ab108319, Abcam, Cambridge, UK), CaMKII (1:1000; #11945, CST), p-CaMKII T286 (1:1000; #12716, CST), CASP3 (1:1000; #9661, CST), CREB (1:500; #sc-186, Santa Cruz Biotechnology, Dallas, TX, USA), p-CREB S133 (1:1000; #06-519, Millipore, Burlington, MA, USA), ERK (1:1000; #9102, CST), p-ERK T202/Y204 (1:1000; #9101, CST), RSK (1:1000; #9355, CST), p-RSK S380 (1:1000; #11989, CST), TRKB (1:500; #4603, CST), p-TRKB Y516 (1:800; #4619, CST), p-TRKB Y817 (1:1000; #ABN1381, Millipore), or GAPDH (1:1000; #30000002, MDBio, Taipei, Taiwan). The secondary antibodies used were horseradish peroxidase-conjugated goat anti-mouse (#GTX213111-01) or goat anti-rabbit (#GTX213110-01) IgG (1:5000; GeneTex, Irvine, CA, USA). Protein signals were detected with chemiluminescent substrate (Millipore) using the ImageQuant™ LAS 4000 imager and analysis software (GE Healthcare, UK).

### 2.9. RNA Interference

Knockdown of TRKB was performed using TRKB-specific lentiviral short hairpin RNA (shRNA) TRCN0000002243, TRCN0000002245 and TRCN0000002246, and a scrambled negative control (TRC2.Void) (RNAi core facility of Academia Sinica, Taipei, Taiwan), as previously described [34]. At day 2 after cell seeding, multiplicity of infection of 3 for each shRNA was used to infect cells with the existence of polybrene (8 μg/mL; Sigma-Aldrich). The cultured media were changed at day 3, following by 8 h compound (10 μM) pre-treatment and doxycycline (2 μg/mL) induction of ΔK280 Tau_RD_-DsRed expression. Cells were collected at day 9 for TRKB protein analysis or imaged for neurite outgrowth analysis as described above.

### 2.10. Tryptophan Fluorescence Quenching Assay

The interaction between the N-terminal extracellular domain of TRKB (TRKB-ECD) and 7,8-DHF (as positive control; [35]), ZN-015, VB-030 or VB-037 was evaluated with tryptophan fluorescence titration. TRKB-ECD-His protein was expressed in the *Pichia* expression system (Invitrogen) and purified as stated [8]. Intrinsic tryptophan fluorescence spectra with titration of a different concentration of compounds (0–1000 nM) were recorded by fluorescence spectrophotometer F-7000 (Hitachi, Tokyo, Japan), with excitation wavelength 295 nm and emission wavelength in the range of 300–400 nm. The calculation of dissociation constant (*K_D_*) between the 7,8-DHF or ZN/VB compound and TRKB was described in a previous report [8].

### 2.11. Statistical Analysis

Data are presented as mean ± standard deviation. Statistical analysis of data was evaluated using the two-tailed Student’s *t*-test or one-way analysis of variance (ANOVA) with Tukey’s *post hoc* adjustment. Significance was considered as a *p*-value < 0.05.

## 3. Results

### 3.1. Cytotoxicity of Heterocyclic ZN/VB Compounds

Flavone ZN-006, benzofuran ZN-013, coumarins ZN-014 and ZN-015, and quinolines VB-030, VB-031, VB-037 and VB-041 were studied (Figure 1A). MTT assay on uninduced ΔK280 Tau_RD_-DsRed SH-SY5Y cells after treatment with test compounds for 24 h showed IC_50_ values greater than 100 μM, except for ZN-015 (77 μM) and VB-031 (6 μM) (Figure 1B). The above results denote that most of these heterocyclic compounds have low cytotoxicity.

### 3.2. Screening Compounds Promoting ΔK280 Tau_RD_-DsRed Folding and Reducing Cellular Oxidative Stress

To unravel whether these compounds promote proper folding of pro-aggregant Tau (ΔK280 Tau_RD_), a high-content fluorescence assay of neuronally differentiated and compound-treated ΔK280 Tau_RD_-DsRed-expressed SH-SY5Y cells [7] was conducted (Figure 2A). Congo red (the positive control) at 10 μM raised DsRed fluorescence to 114% (*p* < 0.001). Among 8 tested compounds, ZN-015 (5–10 μM), VB-030 (10 μM) and VB-037 (5–10 μM) significantly increased DsRed-fluorescence (110–121%, *p* = 0.001−< 0.001) (Figure 2B). Based on the cell number counted, IC_50_ values of congo red, ZN-015, VB-030 and VB-037 were ≫10, 16, ≫10, and 23 μM respectively, in SH-SY5Y cells expressing ΔK280 Tau_RD_-DsRed for 6 days. When the expressed Tau_RD_-DsRed mRNA was quantitated by RT-PCR, no significant fold change between untreated and compound-treated groups was observed (24.2 vs. 24.2–25.5, *p* > 0.05; Figure 2C). As the expressed ΔK280 Tau_RD_-DsRed could lead to accumulation of oxidative stress [7], we examined the cellular ROS level in compound-treated cells using cell-permeant ROS indicator H_2_DCFDA. As Figure 2D shows, treatment of congo red, ZN-015 and VB-030 counteracted the increase of ROS (from 123% to 104–100%, *p* = 0.033–0.015). Overall, the results show that ZN-015 and VB-030 promote ΔK280 Tau_RD_-DsRed protein-folding and reduced cellular ROS level. Although ΔK280 Tau_RD_-DsRed promoted protein-folding, VB-037 had no effect on reducing ROS.

### 3.3. Chemical Chaperone and Antioxidant Activities of ZN/VB Compounds

Next, we examined the effects of congo red and ZN/VB compounds on Tau misfolding using His-tagged ΔK280 Tau_RD_ proteins purified from *E. coli* cells [33] in thioflavin T spectroscopic assay. As shown in Figure 3A, after 48 h incubation at 37 °C, markedly increased thioflavin T fluorescence was observed with ΔK280 Tau_RD_ protein (from 193 to 12,215 arbitrary unit (AU); *p* < 0.001), and the increase was significant compared to wild-type Tau_RD_ protein (12,215 versus 3202 AU; *p* < 0.001). ΔK280 Tau_RD_ aggregation was significantly reduced by congo red (20 μM), ZN-015 (10–20 μM), VB-030 (20 μM) and VB-037 (10–20 μM) (from 12,215 to 6370–3985 AU; *p* = 0.027−< 0.001). EC_50_ values inhibiting ΔK280 Tau_RD_ aggregation were 13.9, 13.2, 20.3 and 13.1 μM for congo red, ZN-015, VB-030 and VB-037, respectively. Additionally, DPPH and ORAC assays were performed to evaluate the antioxidant activity of ZN/VB compounds. Compared to positive-control kaempferol, only VB-037 has very weak DPPH free radical scavenging activity, while ZN-015 in 4–100 μM displayed trolox equivalent activity of 4.6–41.1 μM (Figure 3B). In summary, the results of chemical chaperone activity of ZN-015, VB-030 and VB-037 may explain the increase of DsRed fluorescence, and the oxygen radical scavenging activity of ZN-015 may reflect the reduction of ROS in ΔK280 Tau_RD_-expressing cells.

### 3.4. Neuroprotective Effects of ZN/VB Compounds

The linkage between pathologically modified Tau and Tau aggregates in neurodegeneration has been widely addressed [36]. The activity of caspase-6 protease, associated with axonal degeneration [37], is positively regulated by caspase-1 [38]. Therefore, we assessed the neuroprotective effects of the tested ZN/VB compounds by examining neurite outgrowth and caspase-1/6 activity using congo red as a positive control. As shown in Figure 4A, ZN-015 displayed the best effects on promoting neurite length (from 30.6 to 40.7 μm, *p* < 0.001), process (from 2.07 to 2.34, *p* = 0.044) and branch (from 0.74 to 0.92, *p* < 0.001). Treatment of congo red, VB-030 or VB-037 also resulted in improved neurite length (36.2–36.4 μm, *p* = 0.027–0.019) and branch (0.85–0.86, *p* = 0.020–0.016). Furthermore, caspase-1 activity increased (111.4%, *p* = 0.011) upon induction of ΔK280 Tau_RD_-DsRed expression, and treatment with congo red, ZN-015, VB-030 or VB-037 counteracted the effect (102.3–95.2%, *p* = 0.048−< 0.001). The expressed ΔK280 Tau_RD_-DsRed protein also raised caspase-6 activity (110.5%, *p* = 0.049), but only ZN-015 treatment showed significant inhibition (99.8%, *p* = 0.042) (Figure 4B). In addition, since new generation of AChE inhibitors targeting in Alzheimer’s-associated memory impairment are currently being inspected in human clinical trials [39], we examined whether ZN/VB compounds may reduce AChE activity (Figure 4C). Although induction of ΔK280 Tau_RD_-DsRed expression did not elevate AChE activity, treatment with ZN-015 and VB-030 significantly suppressed endogenous AChE activity (from 90% to 66–59%, *p* = 0.040–0.007). Taken together, the results of neurite outgrowth and caspase 1 activity revealed the neuroprotective potentials of ZN-015, VB-030 and VB-037.

### 3.5. Targets of ZN/VB Compounds on TRKB Pathway

We next examined the therapeutic targets in TRKB signaling for neuroprotection of ZN/VB compounds. LM-031, a coumarin derivative displaying neuroprotective potential by up-regulating CREB-dependent BDNF/BCL2 pathway in pro-aggregatory Tau SH-SY5Y cells [33], was included for comparison. As shown in Figure 5, auto-phosphorylation of TRKB at residues Y516 and Y817 decreased upon the induction of ΔK280 Tau_RD_-DsRed (Y516: 44.4%, *p* = 0.002; Y817: 55.5%, *p* = 0.037). Treatment of LM-031 and ZN-015 significantly upregulated p-TRKB Y817 (115.2–128.3%, *p* = 0.005–0.001), although the level of p-TRKB Y516 did not reach significance (75.6–77.1%, *p* > 0.05). In addition, VB-030 and VB-037 significantly upregulated both p-TRKB Y516 (101.4–115.1%, *p* = 0.002−< 0.001) and Y817 (128.7–154.3%, *p* = 0.001−< 0.001) levels. Figure 5 also showed the downstream kinases activated by these compounds. Treatment of LM-031, ZN-015, VB-030 and VB-037 significantly increased p-AKT S473 (from 73.7% to 116.9–217.6%, *p* = 0.014−< 0.001), p-CaMKII T286 (from 48.9% to 87.6–122.0%, *p* = 0.012−< 0.001), p-ERK T202/Y204 (from 73.8% to 123.1–209.3%, *p* = 0.023−< 0.001) and p-RSK S380 (from 68.3% to 116.7–201.6%, *p* = 0.002−< 0.001). Subsequently, levels of p-CREB S133 (from 73.9% to 96.0–218.6%, *p* = 0.044−< 0.001) and downstream neurotrophic factor BDNF (pro- and mature forms, from 65.6–69.5% to 105.2–147.8%, *p* = 0.045−< 0.001) and anti-apoptotic molecule BCL2 (from 61.2% to 91.4–117.5%, *p* = 0.024−< 0.001) were increased, accompanied by decreased pro-apoptotic protein BAX (from 149.7% to 129.2–113.2%, *p* = 0.043−< 0.001) and apoptosis executor CASP3 (from 139.7% to 105.8–74.7%, *p* = 0.087–0.001). These results indicate that ZN-015, VB-030 and VB-037 could activate TRKB and downstream AKT, CaMKII, ERK and RSK kinases to affect CREB signaling for promotion of neuronal survival.

### 3.6. Effects of TRKB Knockdown on Neurite Outgrowth

To confirm the neuroprotective effects of LM-031, ZN-015, VB-030 and VB-037 through TRKB signaling, lentiviral shRNA specific to TRKB was applied to the ΔK280 Tau_RD_-DsRed-expressed cells (Figure 6A). As Figure 6B showed, expression of TRKB was remarkably knocked down (from 75.9% to 20.2%, *p* < 0.001). Although addition of compounds did not affect the levels of TRKB (75.9% vs. 85.9–50.6%, *p* > 0.05), TRKB-specific shRNA significantly knocked down TRKB expression in these compounds-treated cells (17.6–8.7%, *p* = 0.002−< 0.001). Neurite outgrowth analysis revealed that the improved neurite length (from 29.6 to 37.5–42.4 μm, *p* <0.001) and branch (from 0.62 to 0.87–1.02, *p* = 0.009−< 0.001) by LM-031, ZN-015, VB-030 and VB-037 were restrained by TRKB knockdown (length: 32.7–36.5 μm, *p* = 0.007−< 0.001; branch: 0.74–0.82, *p* = 0.023−< 0.001). For the process of neurite, knockdown of TRKB also effectively counteracted the promoting effect of ZN-015 (from 2.35 to 2.17, *p* < 0.001) (Figure 6C). These results support the neuroprotective effects of LM-031, ZN-015, VB-030 and VB-037 mediated through TRKB signaling.

### 3.7. Binding Affinity of ZN-015, VB-030 and VB-037 with TRKB-ECD

As ZN-015, VB-030 and VB-037 increased TRKB Y516/Y817 auto-phosphorylation, we examined their binding affinity to TRKB. The 398-amino-acid extracellular domain of TRKB (TRKB-ECD, Figure 7A) was expressed in yeast *Pichia pastoris* [8] and used to assess the binding of tested compounds to TRKB-ECD by tryptophan fluorescence assay. A reported TRKB receptor agonist 7,8-DHF [40] was included for comparison. Upon the binding of ligand, the changed TRKB-ECD conformation affects tryptophan microenvironment to result in quench of intrinsic tryptophan fluorescence [35]. The tryptophan fluorescence changes of TRKB-ECD in the presence of test compounds (1–1000 nM) were recorded after excitation at 295 nm at room temperature. Similar to 7,8-DHF, ZN-015, VB-030 and VB-037 displayed decreasing fluorescence in a concentration-dependent manner, and the fluorescence quenching was maximized at 1000 nM of test compounds (Figure 7B). Based on the quantitative analysis of fluorescence change, binding affinity *K_D_* of each compound was calculated (Figure 7B). As a positive control, TRKB-ECD binding affinity of 7,8-DHF was 13.4 ± 8.4 nM, which was closed to the reported value (12.1 ± 1.6 nM, [35]). The observed *K_D_* of 12.2 ± 9.4 nM (ZN-015), 11.0 ± 2.4 nM (VB-030) and 4.3 ± 5.2 nM (VB-037) demonstrated the high binding affinities of selected ZN/VB compounds to the extracellular domain of TRKB receptor. The unexpected raised-fluorescence spectra of VB-030 (500–1000 nM) in the range from 360 to 400 nm at 295 nm excitation was due to the fluorescence self-absorption of the compound itself (Figure 7C).

## 4. Discussion

Given that aberrant aggregation of the Tau protein is one of the pathological hallmarks of AD, treatments targeting tauopathy-induced neurodegeneration may be also beneficial to AD. ΔK280 mutation in the Tau gene influences Tau mRNA splicing and results in a pro-aggregator Tau protein [41]. Similarly, our previous and present studies have shown that ΔK280 mutation induces Tau aggregation, oxidative stress, cytotoxicity, and impaired TRKB signaling [8,34]. The low cytotoxicity of the tested heterocyclic compounds ZN-006 (flavone), -013 (benzofuran), -014 and -015 (coumarin), and VB-030, -037 and -041 (quinoline) suggests their potential as candidate therapeutic compounds for neurodegenerative diseases. We used the established ΔK280 Tau mutation model to show that ZN-015, VB-030 and VB-037 significantly increased DsRed fluorescence, indicating their ability to reduce Tau aggregation. ZN-015, VB-030 and VB-037 reduced Tau fluorescence in the thioflavin T binding assay, indicating their chemical chaperon activity. ZN-015 and VB-030 further decreased ROS in ΔK280 Tau_RD_-DsRed-expressing SH-SY5Y cells. Only ZN-015 displayed an oxygen radical absorbance capacity in biochemical antioxidant assays. Furthermore, ZN-015, VB-030 and VB-037 promoted neurite outgrowth and reduced caspase-1 activity, ZN-015 decreased caspase-6 activity, and ZN-015 and VB-030 lowered AChE activity in ΔK280 Tau_RD_-DsRed-expressing SH-SY5Y cells. Taken together, these results indicate that ZN-015 and VB-030 possess aggregation-inhibitory and anti-oxidative features to provide protection against tauopathy-mediated cytotoxicity. Although not reducing ROS in ΔK280 Tau_RD_-DsRed-expressing SH-SY5Y cells, the antioxidant effect of VB-037 has been demonstrated in cell models for Parkinson’s disease [42]. This neuroprotective effect may be mediated by hormetic responses through the activation of nuclear factor erythroid-derived 2-like 2 antioxidant response elements or heat shock protein-mediated signaling pathways [43,44,45]. Further research is needed to explore the relationship between these signaling pathways and our candidate compounds.

A previous study has shown that over-expression of the wild-type or mutant Tau protein down-regulates BDNF expression in cellular and animal models of AD [9], which was also demonstrated by the present study. Impaired TRKB signaling contributes to the neurodegeneration of tauopathy or AD. Indeed, TRKB reduction exaggerates cognitive impairments and signal dysfunctions in 5XFAD mice [46]. In accordance with the previous report, our study shows that mutant ΔK280 Tau impairs TRKB signaling and neurite outgrowth. Compounds ZN-015, VB-030 and VB-037 activate p-TRKB, p-AKT, p-CaMKII, p-ERK, p-RSK, and p-CREB, elevate BDNF and BCL2, and decrease BAX and CASP3 expression in ΔK280 Tau_RD_-DsRed-expressing SH-SY5Y cells to rescue neurite outgrowth deficits. These results suggest these compounds may act as TRKB agonists to achieve neuroprotection effects. The neurite outgrowth promotion effects are attenuated by the knockdown of TRKB, which provides evidence that these compounds exert their beneficial effects through activating the TRKB signaling pathway. Furthermore, the tryptophan fluorescence assay demonstrates high binding affinities of ZN-015, VB-030 and VB-037 to TRKB-ECD, which are compatible with that of the reported TRKB agonist, 7,8-DHF [26,27].

In this study, we have shown that all of these pathological changes induced by mutant ΔK280 Tau can be rescued by ZN-015, VB-030 and VB-037 via activating TRKB and its downstream signaling, including PI3K/AKT, ERK/CREB, ERK/RSK, PLC-γ/CaMKII, PLC-γ/CREB, and CREB/BDNF pathways [13]. Indeed, these pathways play an important role in neuronal survival, and impairment of them causes AD pathology and cognitive decline. The PI3K/AKT pathway plays a crucial role in regulating cell survival, proliferation, differentiation, intracellular trafficking, and neurite outgrowth [47]. It has been shown that Aβ oligomers attenuate PI3K/AKT signaling significantly and activation of the PI3K/AKT pathway may promote neuronal survival [48]. A previous study has shown that Tau overexpression induced by extrasynaptic N-methyl-D-aspartate (NMDA) receptor activation causes neuronal death through suppressing survival signaling ERK phosphorylation [49]. Furthermore, inhibition of the compensatory increases of the BDNF-ERK-CREB pathway and exacerbates cognitive impairment in vascular dementia associated with obesity [50]. Deficiency of CREB signaling underlies cognitive deficits in aging and also contributes to neurodegeneration in AD, and activation of CREB may be a potential therapeutic strategy in dementia [18,51,52]. PLC-γ/CaMKII promotes intracellular Ca^2+^ release and activates long-term potentiation and synaptic plasticity [53]. In addition, Ca^2+^ further activates CaMKII to phosphorylate CREB and subsequently induces BDNF transcription [54]. RSK is an effector of ERK, and ERK and RSK cooperate in regulation of several proteins, including Fos proto-oncogene, AP-1 transcription factor subunit (FOS), GSK-3β, eukaryotic elongation factor 2 (eEF2) kinase, tuberin and eukaryotic translation initiation factor 4B (eIF4B), to stimulate cell survival [15]. Taken together with our study findings, activating TRKB downstream pathways provides neuroprotection in tauopathy or AD models. Although we have shown that compounds ZN-015, VB-030 and VB-037 protect differentiated SH-SY5Y cells via ameliorating Tau aggregation-related disturbance in the TRKB signal pathway, we did not provide direct evidence of activating TRKB by the tested compounds. Future studies showing direct binding of these compounds to the TRKB are necessary.

Although over-expression of BDNF improves cognitive deficits in AD animal models [21,22], bioavailability and blood–brain barrier permeability of BDNF is poor, and it is invasive and not practical to inject the BDNF-expressing lentivirus into the human brain. Therefore, compounds working as TRKB agonists have much more potential in terms of clinical treatments. Kwon et al. found that plant *Zizyphus jujuba* var. *spinosa* works as a TRKB agonist to attenuate Aβ-induced synaptic long-term potentiation deficits [55]. In addition to 7,8-DHF, Chen et al. also showed that a synthetic derivative CF_3_CN binds with TRKB, activates TRKB signaling, reduces AD pathologies, and ameliorates cognitive dysfunctions in 5XFAD mice [56]. Recently, Wang et al. developed a TRKB agnost antibody that can penetrate into the brain, prevent Aβ-induced cell death, and rescue memory deficits in the APP/PS1 mouse model [57]. We have previously shown that quercetin, apigenin and coumarin derivatives LMDS-1 and -2 activate TRKB to reduce Tau aggregation and protect cells against Tau-induced neurotoxicity [8,34]. In this study we have shown that coumarin derivative ZN-015 and quinoline compounds VB-030 and VB-037 also act as TRKB agonists to protect SH-SY5Y cells from cytotoxicity induced by mutant ΔK280 Tau. However, these results should be confirmed by future animal studies.

## 5. Conclusions

In conclusion, we demonstrated that ZN-015, VB-030 and VB-037 have potentials in anti-aggregation and promoting neuronal survival in pro-aggregator Tau-expressing SH-SY5Y cells. In addition, our results revealed that ZN-015, VB-030 and VB-037 can bind to TRKB-ECD to activate TRKB signaling through AKT, CaMKII, ERK and RSK kinases, all of which promote neurite outgrowth. These multi-functional molecules provide more diversity in pursuance of finding an efficient therapeutic strategy for neurodegenerative diseases such as AD.

## Figures and Tables

**Figure 1 biomolecules-13-00219-f001:**
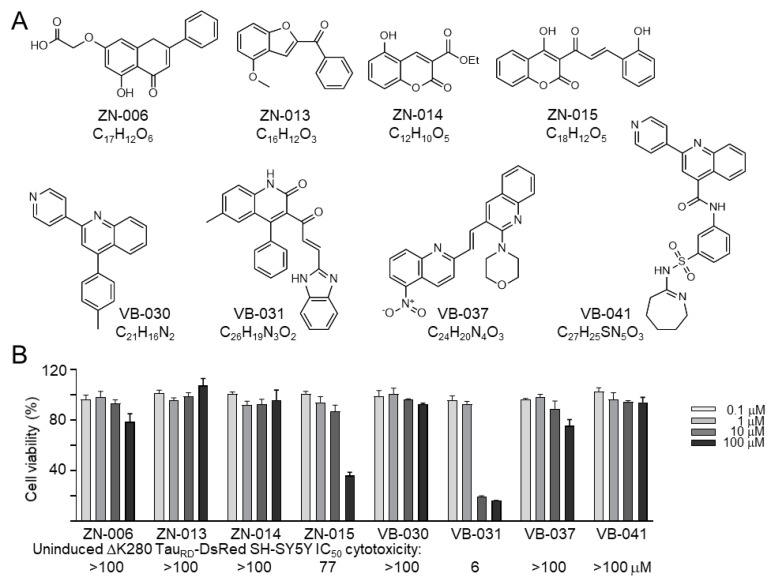
Test compounds. (**A**) Structure and formula of ZN-006, -013, -014, -015, and VB-030, -031, -037, -041. (**B**) MTT cell viability assay of ZN/VB compounds in ΔK280 Tau_RD_-DsRed SH-SY5Y cells, with test compounds (0.1–100 μM) treated for 24 h (*n* = 3). The viability of untreated cells was set at 100% for normalization. Calculated IC_50_ values are shown below.

**Figure 2 biomolecules-13-00219-f002:**
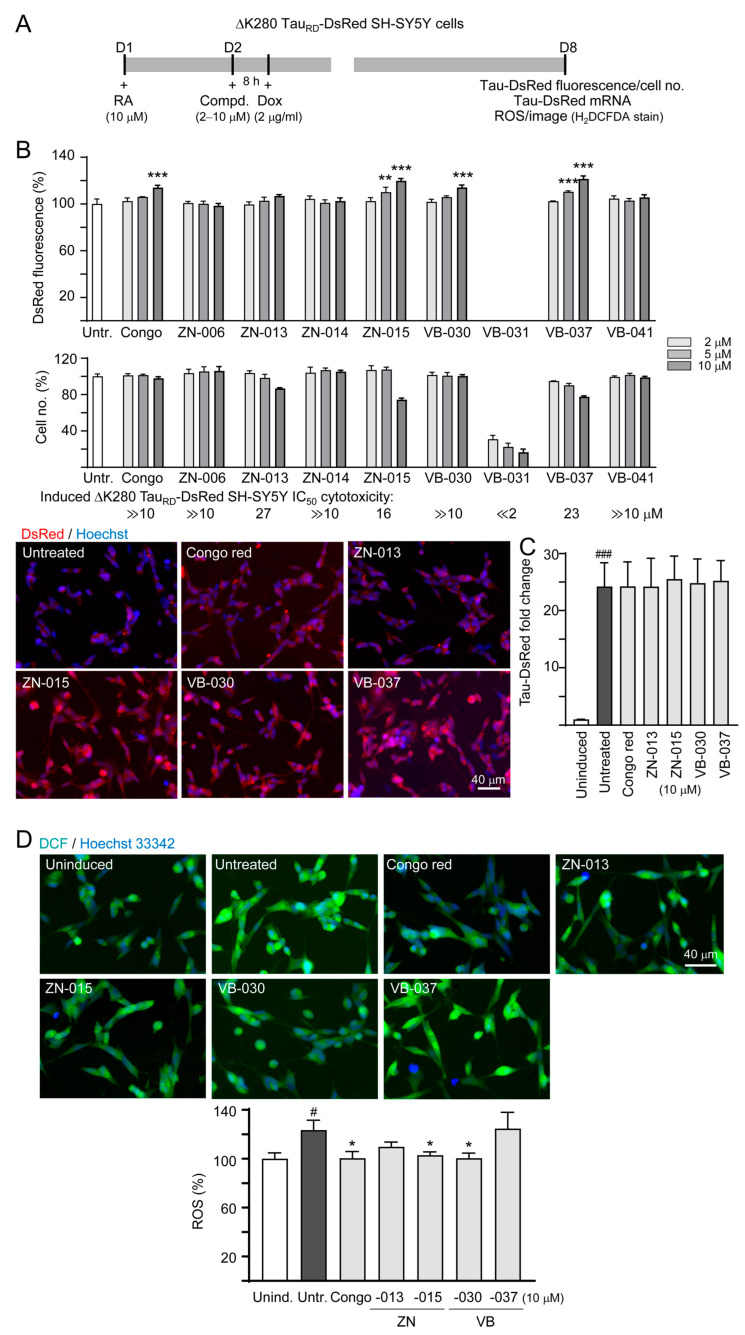
ZN/VB compounds screen using the Tet-On ∆K280 Tau_RD_-DsRed SH-SY5Y cell model for AD. (**A**) Experimental flowchart. At day 1, cells were plated in the presence of retinoic acid (RA, 10 µM). At day 2, cells received an 8 h congo red or ZN/VB compound pre-treatment (2–10 µM), followed by a 6 day ΔK280 Tau_RD_-DsRed induction by doxycycline (Dox, 2 µg/mL). At day 8, DsRed fluorescence, Tau_RD_-DsRed RNA and ROS (H_2_DCFDA stain) were measured. (**B**) High-content assessment of DsRed fluorescence (*n* = 3). The relative DsRed fluorescence of untreated cells was normalized as 100%. Shown below were percentage of cell survival in each treatment. Due to less than 75% cells remained, DsRed fluorescence of VB-031-treated cells was not recorded. In addition, DsRed images of ΔK280 Tau_RD_-DsRed cells without or with congo red, ZN-013, ZN-015, VB-030 or VB-037 (10 µM) treatment were shown. Nuclei were counterstained with Hoechst 33342 (blue). (**C**) Tau_RD_-DsRed RNA fold change in cells with 10 µM congo red or ZN/VB compound treatment (*n* = 3). (**D**) High-content assessment of ROS (DCF fluorescence, green) (*n* = 3). The relative ROS of uninduced cells was normalized (100%). Nuclei were counterstained with Hoechst 33342 (blue). *p* values: comparisons between induced (Dox+) vs. uninduced (Dox-) cells (#: *p* < 0.05, ###: *p* < 0.001), or compound-treated vs. untreated (both are under Dox+) cells (*: *p* < 0.05, **: *p* < 0.01, ***: *p* < 0.001) (one-way ANOVA with a *post hoc* Tukey test).

**Figure 3 biomolecules-13-00219-f003:**
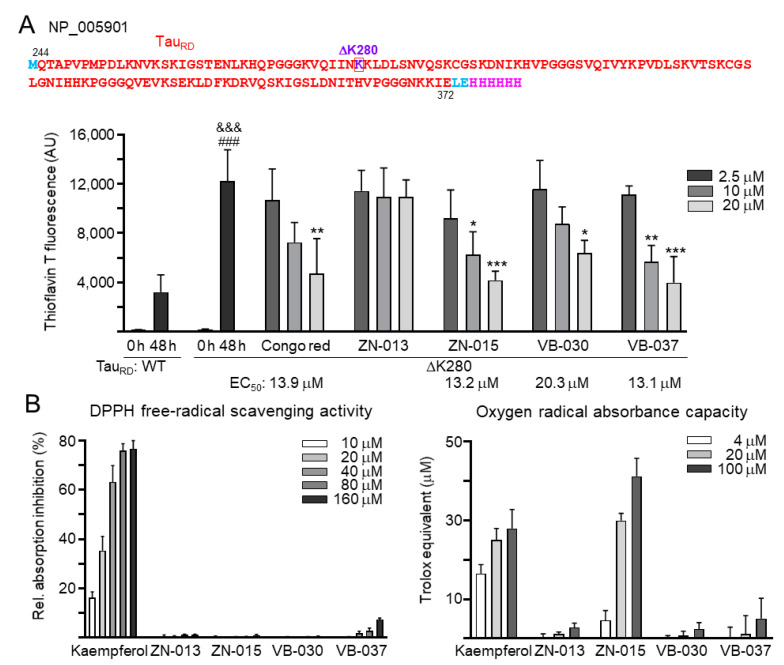
Chemical chaperone and antioxidant activities of ZN/VB compounds. (**A**) Thioflavin T binding assay for Tau_RD_ (Q^244^-E^372^ of Tau, NP_005901) aggregation. Top: Amino acid sequence of Tau_RD_-His protein, with Tau_RD_ marked in red, ΔK280 marked in purple, in-frame-fused His_6_ marked in pink, and N-terminal Met and linker (Leu and Glu) marked in blue. Bottom: Anti-aggregation assay. Tau_RD_ protein (20 μM) was incubated with congo red, ZN-013, ZN-015, VB-030 or VB-037 (2.5–20 μM) at 37°C for 48 h, and aggregation was monitored by measuring thioflavin T fluorescence intensity (*n* = 3). *p* values: comparisons between 0 h and 48 h (^###^: *p* < 0.001), wild type versus ΔK280 (^&&&^: *p* < 0.001), or with and without compound addition (*: *p* < 0.05, **: *p* < 0.01, ***: *p* < 0.001) (one-way ANOVA with a *post hoc* Tukey test). (**B**) Antioxidant assays of kaempferol (as a positive control) and ZN/VB compounds (10–160 μM for DPPH free radical scavenging activity and 4–100 μM for trolox equivalent oxygen radical absorbance capacity) (*n* = 3).

**Figure 4 biomolecules-13-00219-f004:**
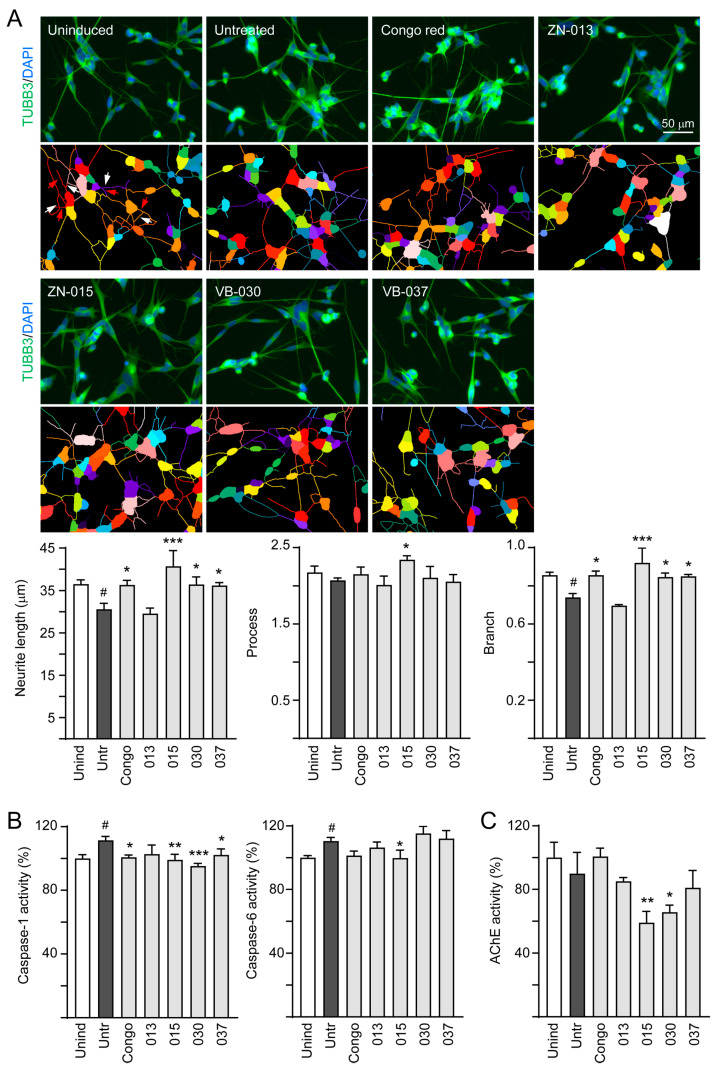
Neuroprotective effects of ZN/VB compounds in ΔK280 Tau_RD_-DsRed SH-SY5Y cells. (**A**) High-content assessment of neurite outgrowth (TUBB3 stain, green) (*n* = 3). The relative length, process, and branch of neurite of uninduced cells was normalized (100%). Nuclei were counterstained with DAPI (blue). The multi-colored mask was applied on segmented fluorescence images to assign the outgrowth of a cell body for quantification. Different colors outline different neurons and their respective neurite outgrowth. The arrows indicated in the uninduced photo are process (red) and branch (white). The total length, process and branch of neurite were calculated. (**B**) Caspase-1, caspase-6, and (**C**) AChE activity assessments (*n* = 3). The relative caspase-1/caspase-6/AChE activity of uninduced cells (Dox-) was normalized (100%). *p* values: comparisons between induced (Dox+) vs. uninduced (Dox-) cells (^#^: *p* < 0.05), or compound-treated vs. untreated (induced) cells (*: *p* < 0.05, **: *p* < 0.01, ***: *p* < 0.001) (one-way ANOVA with a *post hoc* Tukey test).

**Figure 5 biomolecules-13-00219-f005:**
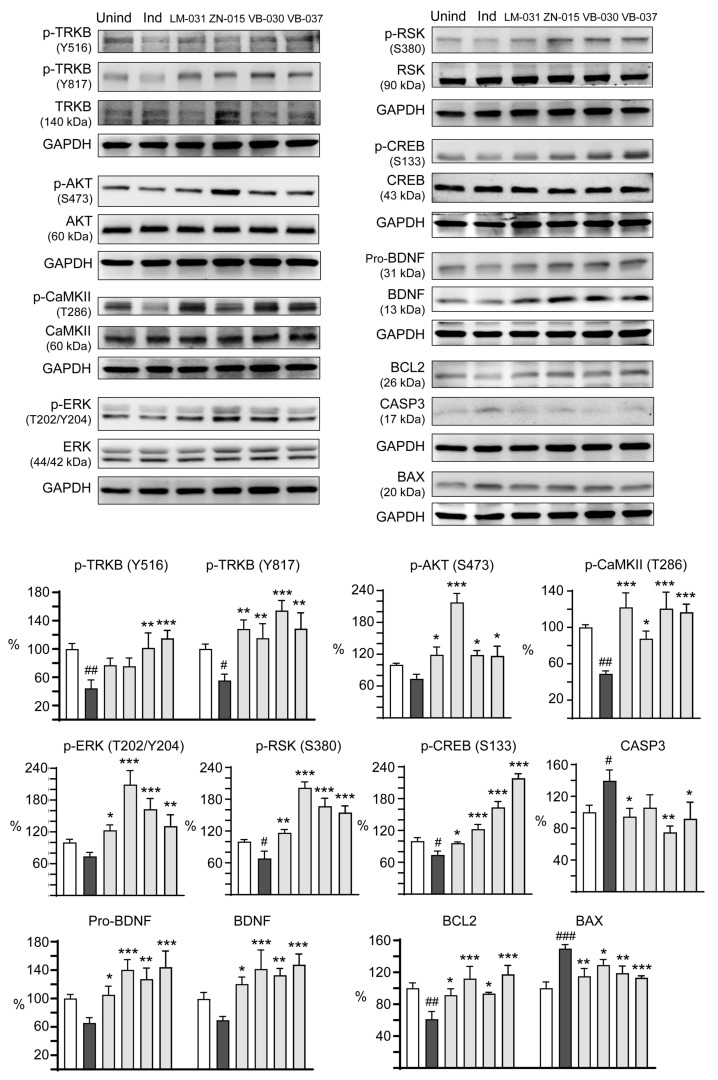
Activation of TRKB signaling in ∆K280 Tau_RD_-DsRed SH-SY5Y cells. p-TRKB (Y516 and Y817), TRKB, p-AKT (S473), AKT, p-CaMKII (T286), CaMKII, p-ERK (T202/Y204), ERK, p-RSK (S380), RSK, p-CREB (S133), CREB, BDNF (pro- and mature), BCL2, BAX and CASP3 levels analysed by immunoblotting using GAPDH as a loading control (*n* = 3). To normalize, the protein expression level in uninduced cells was set at 100%. *p* values: comparisons between induced vs. uninduced cells (^#^: *p* < 0.05, ^##^: *p* < 0.01, ^###^: *p* < 0.001), or compound-treated vs. untreated cells (*: *p* < 0.05, **: *p* < 0.01, ***: *p* < 0.001) (one-way ANOVA with a *post hoc* Tukey test).

**Figure 6 biomolecules-13-00219-f006:**
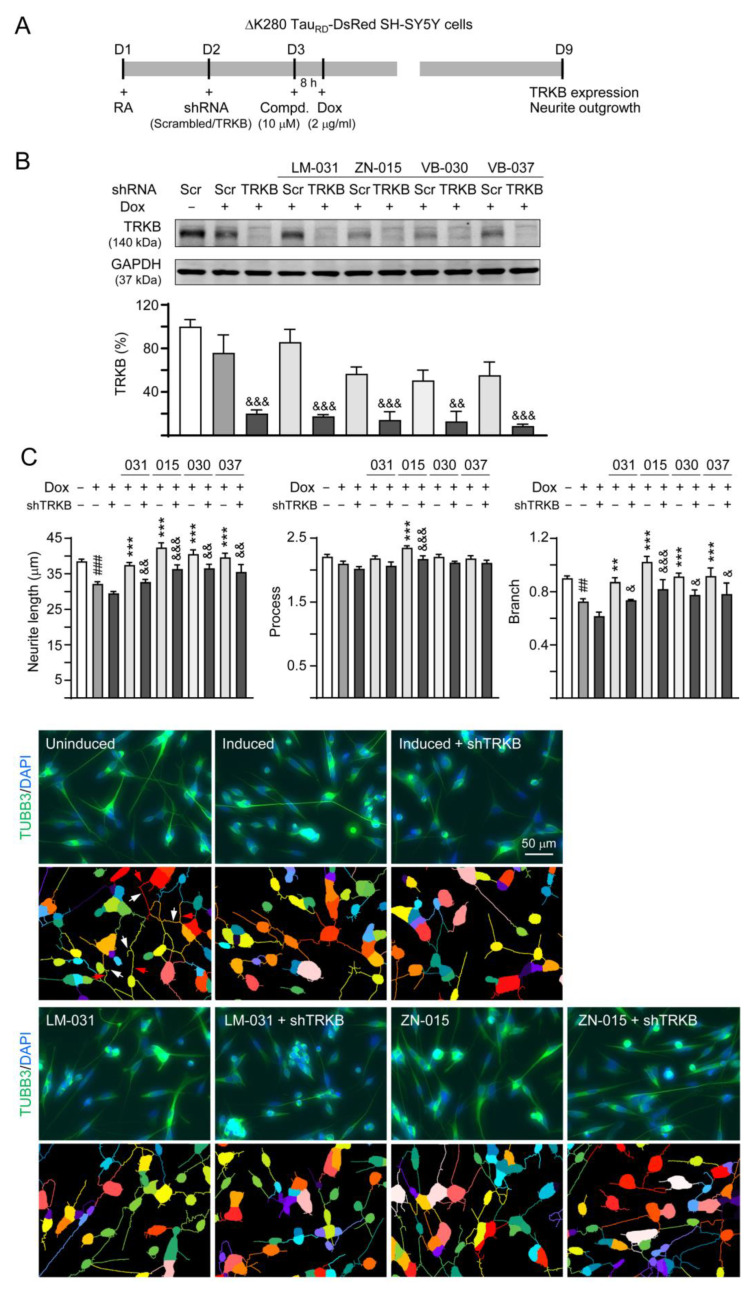
TRKB RNA interference of ∆K280 Tau_RD_-DsRed SH-SY5Y cells. (**A**) Experimental flow chart. On day 1, ΔK280 Tau_RD_-DsRed SH-SY5Y cells were plated in the presence of retinoic acid (RA; 10 µM). The cells were infected with lentivirus-expressing TRKB-specific or scrambled shRNA at day-2. At 24 h post-infection, LM-031, ZN-015, VB-030 or VB-037 (10 µM) was added for 8 h, followed by induction of Tau_RD_-DsRed expression (Dox, 2 µg/mL) for 6 days. The cells were collected for TRKB and neurite outgrowth analyses at day 9. (**B**) Protein level of TRKB in compound-treated cells infected with TRKB-specific or scrambled shRNA-expressing lentivirus (*n* = 3). GAPDH was used as a loading control. The relative TRKB of uninduced cells was normalized to 100%. (**C**) High-content assessment of neurite outgrowth (TUBB3 stain, green) (*n* = 3). The length, process and branch of neurite was calculated. Nuclei were counterstained with DAPI (blue). Shown below are the multi-color segmented fluorescence images to assign the outgrowth of a cell body for quantification. Different colors outline different neurons and their respective neurite outgrowth. The arrows indicated in uninduced photo are process (red) and branch (white). *p* values: comparisons between induced vs. uninduced cells (^##^: *p* < 0.01, ^###^: *p* < 0.001), compound-treated vs. untreated (induced) (**: *p* < 0.01, ***: *p* < 0.001), or TRKB shRNA-treated vs. scrambled shRNA-treated cells (^&^: *p* < 0.05, ^&&^: *p* < 0.01, ^&&&^: *p* < 0.001) (one-way ANOVA with a *post hoc* Tukey test).

**Figure 7 biomolecules-13-00219-f007:**
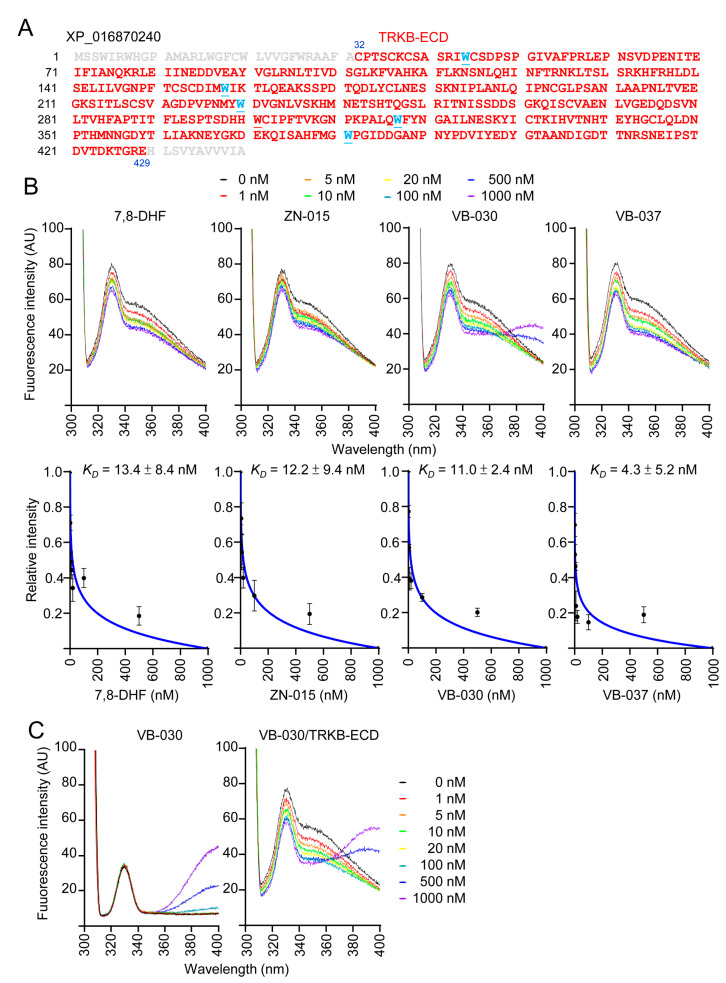
Binding assay of ZN/VB compounds to the extracellular domain of the TRKB. (**A**) Amino acid sequences of TRKB-ECD (marked in red, C^32^-E^429^, XP_016870240), with tryptophan (W) underlined and marked in blue. (**B**) Top: Tryptophan fluorescence of TRKB-ECD recombinant protein titrated with 7,8-DHF, ZN-015, VB-030 and VB-037 (0–1000 nM). The tryptophan fluorescence of TRKB-ECD excited at 295 nm showed decreased intensity after titrating with increasing concentrations of test compounds. Bottom: The plot of normalized tryptophan intensities at emission 330 nm as a function of total added concentrations of test compounds. The dissociation constant (*K_D_*) between test compounds and TRKB-ECD was calculated by fitting this plot with the equation described. (**C**) Fluorescence spectra of VB-030 (0–1000 nM) in the absence (left) or presence (right) of TRKB-ECD.

## Data Availability

The data presented in this study are available on request from the corresponding author.

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
