# Peer review of "Using ΔK280 TauRD Folding Reporter Cells to Screen TRKB Agonists as Alzheimer’s Disease Treatment Strategy"

_biomolecules, 2023, doi:10.3390/biom13020219_

Round 1

Reviewer 1 Report

Interplay and coordination of redox interactions with endogenous and exogenous stress-response and antioxidant defence systems  is an emerging area of reserach interest in  anti-inflammatory anti-degenerative therapeutics. In numerous experimental models, natural antioxidants induce hormetic dose responses displaying endpoints of biomedical and clinical relevance. This reviewer is satisfied with the significance of this study, the care in which the study was performed, and the implications of the results for human health.  Results presented are interesting  and the questions posed are of extremely high interest, thus the paper does give adequate definitive information. Pending minor points, this paper can be accepted

Minor concerns:

1. Preconditioning signal leading to cellular protection through Hormesis is a well known protective mechanism against neuroinflammatory damage. Given the relationship between vitagene network and its possible biological relevance in the defense mechanisms against  oxidative stress-driven degenerative diseases, Authors can mention in the discussion appropriately this aspect (See and quote please Calabrese et al., 2010, Antiox. Redox Signal 13,1763; Calabrese et al., Nature Neurosci., 2007 8, 766; Calabrese EJ., et al., 2012 Biogerontology 13, 215).

Reviewer 2 Report

The authors showed that several compounds block intracellular Tau aggregation and recover TrkB-related neuroprotective signal transduction. 

Although the authors claimed that the compounds, such as ZN-015, protect neuronally differentiated SH-SY5Y cells via TrkB, they did not show the direct evidence that the compounds activate TrkB by themselves. The compounds may just ameliorate Tau aggregation-related disturbance in TrkB signal. This reviewer recommends to confirm whether the compounds can activate and induce TrkB signal transduction by using wild-type SH-SY5Y cells.

In Figure 3, additional biochemical analyses would help to confirm the neuroprotective effects of the compounds. Synaptophysin would be a good indicator.

Round 2

Reviewer 2 Report

Although the authors didn't carry out additional experiments recommended by this reviewer, they rearranged the manuscript to avoid misleading readers. The reviewer feels that this revision looks much reasonable.